# 4-Arylthiosemicarbazide Derivatives as Toxoplasmic Aromatic Amino Acid Hydroxylase Inhibitors and Anti-inflammatory Agents

**DOI:** 10.3390/ijms23063213

**Published:** 2022-03-16

**Authors:** Adrian Bekier, Anna Brzostek, Agata Paneth, Bożena Dziadek, Jarosław Dziadek, Justyna Gatkowska, Katarzyna Dzitko

**Affiliations:** 1Department of Molecular Microbiology, Faculty of Biology and Environmental Protection, University of Lodz, 90-237 Lodz, Poland; adrian.bekier@biol.uni.lodz.pl (A.B.); bozena.dziadek@biol.uni.lodz.pl (B.D.); justyna.gatkowska@biol.uni.lodz.pl (J.G.); 2Laboratory of Genetics and Physiology of Mycobacterium, Institute of Medical Biology, Polish Academy of Sciences, 93-232 Lodz, Poland; abrzostek@cbm.pan.pl (A.B.); jdziadek@cbm.pan.pl (J.D.); 3Department of Organic Chemistry, Faculty of Pharmacy, Medical University of Lublin, 20-093 Lublin, Poland; agatapaneth@umlub.pl

**Keywords:** *Toxoplasma gondii*, tyrosine, thiosemicarbazide, bradyzoite, NF-κB pathway inhibition

## Abstract

Approximately one-third of the human population is infected with the intracellular cosmopolitan protozoan *Toxoplasma gondii* (*Tg*), and a specific treatment for this parasite is still needed. Additionally, the increasing resistance of *Tg* to drugs has become a challenge for numerous research centers. The high selectivity of a compound toward the protozoan, along with low cytotoxicity toward the host cells, form the basis for further research, which aims at determining the molecular targets of the active compounds. Thiosemicarbazide derivatives are biologically active organic compounds. Previous studies on the initial preselection of 58 new 4-arylthiosemicarbazide derivatives in terms of their anti-*Tg* activity and selectivity made it possible to select two promising derivatives for further research. One of the important amino acids involved in the proliferation of *Tg* and the formation of parasitophorous vacuoles is tyrosine, which is converted by two unique aromatic amino acid hydroxylases to levodopa. Enzymatic studies with two derivatives (R: *para*-nitro and *meta*-iodo) and recombinant aromatic amino acid hydroxylase (AAHs) obtained in the *E. coli* expression system were performed, and the results indicated that toxoplasmic AAHs are a molecular target for 4-arylthiosemicarbazide derivatives. Moreover, the drug affinity responsive target stability assay also confirmed that the selected compounds bind to AAHs. Additionally, the anti-inflammatory activity of these derivatives was tested using THP1-Blue™ NF-κB reporter cells due to the similarity of the thiosemicarbazide scaffold to thiosemicarbazone, both of which are known NF-κB pathway inhibitors.

## 1. Introduction

Parasites such as *Toxoplasma*, *Plasmodium*, and *Cryptosporidium* of the *Apicomplexa* phylum cause significant morbidity and mortality in humans and in livestock [1,2,3,4]. Together, these parasites cause over 0.5 million deaths each year and enormous financial losses [5,6,7].

Toxoplasmosis is a dangerous parasitosis, especially for developing fetuses, due to the risk of congenital infection. Moreover, *Toxoplasma gondii* (*T. gondii*) invasion may lead to brain damage and even death in immunocompromised individuals. The main route of parasite transmission to humans involves the ingestion of raw meat containing tissue cysts or water or vegetables contaminated with oocysts. Additionally, individuals can become infected horizontally (iatrogenic) via blood transfusion or organ transplantation and vertically from mother to fetus via the placenta [1]. The parasite is also responsible for livestock infections. Farm animals that are bred for human consumption can acquire *T. gondii* infection through ingesting sporulated oocysts in water or plants. After the release of bradyzoites from tissue cysts and sporozoites from oocysts, which takes place in the intestines, both parasites transform into tachyzoites, and this process marks the beginning of the acute phase of parasite invasion. Under the pressure of developed immunity, tachyzoites convert into slow-dividing bradyzoites, which are enclosed within tissue cysts and are localized in various tissues, e.g., neural and/or muscle [1,2,8]. It is worth emphasizing that currently, the only effective means of preventing *T. gondii* infection is preventive health care, such as increasing the awareness of future mothers and achieving early diagnoses in pregnant women and newborns. An efficient method to completely eliminate the parasite from an infected organism has not yet been developed.

Amino acids are essential to survival, as they not only build proteins but are also key molecules that are necessary for metabolic pathways to properly function. The important fact about the biosynthesis and acquisition routes for amino acids in human-infecting parasites is that they must either salvage the amino acids from the host or synthesize them themselves. There are several essential amino acids that must be taken up by *T. gondii*, including arginine (Arg), histidine (His), leucine (Leu), isoleucine (Ile), valine (Val), methionine (Met), phenylalanine (Phe), tryptophan (Trp), and tyrosine (Tyr) [9,10]. The last three of these are aromatic amino acids that are essential to *Toxoplasma* growth and must be salvaged from the host [9,10,11,12,13,14,15,16].

Two nearly identical isoforms of aromatic amino acid hydroxylase (AAH1 and AAH2) can produce levodopa, a precursor for dopamine (a catecholamine neurotransmitter), from Phe and Tyr [14]. It was determined that levodopa is also a component of the oocyst wall, so AAH1 and AAH2 are important at this stage of the life cycle [13]. Although the role of AAH1 and AAH2 proteins in the tachyzoite or bradyzoite stages has not been reported thus far, close examination of tyrosine metabolism at these stages has revealed that they are both dependent on the addition of exogenous Tyr for efficient growth. Additionally, when access to Tyr is limited, it is impossible to efficiently form parasitophorous vacuoles [12].

Moreover, a family of plasma membrane-localized amino acid transporters named apicomplexan amino acid transporters (ApiATs), which are ubiquitous in *T. gondii* and other apicomplexan parasites, was described [11]. Characterization of these transporters in *T. gondii* shows that they are important for intracellular growth at the tachyzoite stage of the parasite, which is related to acute infections. A number of these proteins, including TgApiAT2, TgApiAT3-1, TgApiAT3-2 TgApiAT3-3, and TgApiAT6-3, are described as putative amino acid transporters but are not highly specific. The TgApiAT5-3 protein is the exchanger for aromatic and large neutral amino acids and is particularly important for the Tyr uptake pathway and amino acid homeostasis, which are critical for parasite virulence. Furthermore, tachyzoites lacking this transporter displayed a phenotype with decreased robustness, but it could be rescued if additional aromatic amino acids, such as Tyr, Phe, or Trp, were added to the media. This finding reflects the presence of an alternate amino acid transporter, which primarily transports Phe and Trp, with a lower affinity for Tyr. Additionally, TgApiAT1, TgApiAT2, and TgApiAT5-3 must be present for *T. gondii* to have normal intracellular growth in in vitro culture [11].

The reports in [11,12,13,14,15,17,18,19] indicate the concept that compounds that inhibit exogenous Tyr uptake into *T. gondii* or participate in the disruption of Tyr metabolism in parasites have the potential to be developed into vital medicines for the effective treatment of toxoplasmosis. Unfortunately, to date, there are no effective treatments against bradyzoites, and medicines that target the tachyzoites (pyrimethamine and sulfadiazine are the most effective) are associated with toxicity and hypersensitivity [20,21,22]. Other serious problems, such as drug resistance and relapses after therapy is discontinued, have also been reported [23,24,25,26]. Hence, novel, efficacious anti-toxoplasma agents are greatly needed.

Additionally, thiosemicarbazide derivatives are biologically active, noncytotoxic organic compounds with proven antitumor, antiviral, antifungal, and antioxidant activities [27,28,29,30,31,32] but are also precursors for thiosemicarbazones, which are formed by the condensation of thiosemicarbazide or substituted thiosemicarbazide at the N4 position with a suitable aldehyde or ketone [33]. Last, thiosemicarbazone derivatives were described as anti-inflammatory factors, especially in the inflammatory response mediated by nuclear factor kappa light-chain-enhancer of activated B cells (NF-κB), by inhibiting NF-κB transactivation [33,34,35,36]. *T. gondii* invasion, through some inflammatory stimulants, such as toxoplasmal proteins, activates immune responses through the activation of NF-κB [37]. Our 4-arylthiosemicarbazide derivatives could have dual activity, in which the first major, anti-proliferative effect is on *T. gondii* and the second anti-inflammatory effect occurs by blocking the NF-κB pathway, which is similar to how thiosemicarbazones work.

Our group has developed a series of 4-arylthiosemicarbazide derivatives as candidates to inhibit aromatic amino acid hydroxylases. The presented studies demonstrate our approach to supporting the hypothesis that the biological effect of the examined compounds results from the disruption of *T. gondii* Tyr metabolism by inhibiting AAH. First, the problem of the cytotoxicity of the studied compounds was addressed. Second, we measured the studied compounds’ in vitro anti-*T. gondii* activity. Once the prepared compounds were determined to be nontoxic and biologically active, we investigated whether their effect on *T. gondii* was related to their inhibitory activity against AAH. Additionally, the anti-inflammatory activity of the 4-arylthiosemicarbazide derivatives was assessed using THP1-Blue™ NF-κB reporter cells.

## 2. Results

### 2.1. Rationale

We have recently shown the effects of the analyzed 4-arylthiosemicarbazide derivatives on the inhibition of *T. gondii* growth and their cytotoxicity against mouse (L929) and human (Hs27) fibroblast cell lines (Table 1) [16,38,39]. Both derivatives, **2a** and **6a**, are highly selective for the parasite and possess anti-tyrosinase (anti-TYR) activity, which was described in our previous study [16].

### 2.2. Confirmation of Bradyzoite Differentiation

To confirm differentiation from the tachyzoite to the bradyzoite life cycle stages, protein expression during pH shock-induced stage conversion was monitored. First, differentiation was proven using an immunofluorescent assay with anti-MAG1 and anti-SAG1 rabbit antibodies, but also by scoring changes in parasitemia and the number of parasites per vacuole (data not shown). The expression of the bradyzoite-specific protein MAG1 was observed, and the expression of the tachyzoite-specific gene SAG1 decreased postinduction (Figure 1).

### 2.3. Expressing and Purifying Recombined AAHs

The AAHs in the supernatants of bacterial sonication extraction which exhibited the highest amounts of recombinant AAHs were purified using HisPur™ Cobalt Resin. The AAHs were isolated by SDS–PAGE gel as an individual band with a molecular weight of 40.8 kDa. The results of the Western blot also revealed the recognition of both AAH1 and AAH2 by anti-His-tag and anti-TH antibodies, resulting in bands with a molecular mass of 40.8 kDa (Figure 2).

### 2.4. Biochemical Assays

The screening studies of TYR inhibition highlighted two distinctive 4-arylthiosemicarbazides (**2a** and **6a**) with significant inhibitory activity [16]. Therefore, these compounds were subsequently subjected to detailed kinetic studies to evaluate the inhibition of the AAH1 and AAH2 parameters. The Lineweaver–Burk plots (Figure 3), which are the linearized transformations of the Michaelis–Menten curves, clearly demonstrate the competitive character of the inhibition capabilities of the studied compounds. The values of the kinetic parameters *V_max_/K_M_*, *K_M_*, and *K_I_* were calculated and are presented in Table 2.

### 2.5. Drug Affinity Responsive Target Stability Assay

Incubation with small molecules (**2a** and **6a**) confers specific protection against proteolysis of toxoplasmic aromatic amino acid hydroxylase 1 (AAH1). We also performed this assay with BSA as a control, and the activity of Pronase was not inhibited by our compounds. SDS–PAGE of BSA from control and whole Western blots of AAH1 from the drug affinity responsive target stability assay are presented in Appendix A.

#### 2.5.1. Protection of AAH1 from Proteolysis by 4-Arylthiosemicarbazide Derivatives in the Presence of 10 µM of Compound

Bands that appeared to be protected by incubation with 4-arylthiosemicarbazide derivatives (**2a** and **6a**) compared to the vehicle control (DMSO) were found. The results from the proteolytic experiment are shown in Figure 4. As a proof-of-principle, we examined protein AAH1, which is the target for our derivatives. Western blotting illustrated the presence of AAH1 protein at low Pronase:protein ratios and its reduction and loss with increasing ratios (Figure 4A,B). Proteolysis of AAH1 by Pronase was clearly inhibited by the presence of **2a** and **6a**, and the addition of these two compounds significantly inhibited proteolytic activity (Figure 4C,D).

#### 2.5.2. Protection of AAH1 from Proteolysis by 4-Arylthiosemicarbazide Derivatives in the Presence of Increasing Concentrations of Compounds

To investigate the effects of 4-arylthiosemicarbazide derivative concentrations, we maintained a constant Pronase:protein ratio while varying concentrations of **2a** and **6a**. As the ligand concentration nears target binding saturation, a higher amount of target protein can be observed. 4-Arylthiosemicarbazide derivatives dose-dependently enhanced the level of AAH1, suggesting the stability of AAH1 increases with treatment by **2a** or **6a** (Figure 5A,B). Quantification of the target protein band intensities allows the target stability to be represented as a function of ligand concentration, as exemplified by the curve in Figure 5C,D. These results strongly suggest that AAH1 is the target protein of 4-arylthiosemicarbazide derivatives.

### 2.6. Anti-inflammatory Activity of 4-Arylthiosemicarbazide Derivatives

Using the THP1-Blue™ NF-κB reporter cell line, we showed that 4-arylthiosemicarbazide derivatives possess anti-inflammatory activity. All toxoplasmic stimulants and commercially available LPS induce NF-κB activation in these cells. The LPS-induced activation was similar to that of AAH1 and AAH2-treated cells but higher than that of TLA- and BLA-treated cells. Cells were treated with the **2a** or **6a** derivatives 2 h after or before stimulation. Interestingly, both derivatives significantly reduced monocyte activation after stimulation when compared with no treatment (*p* < 0.0001) (Figure 6).

Additionally, the level of INF-γ production in the supernatant after human monocytes were exposed to TLA, BLA, AAH1, or AAH2 or were treated was measured. According to the data obtained from NF-κB activation, only monocytes that were stimulated but not treated showed a significant increase in INF-γ production (Figure 7).

## 3. Discussion

Herein, we described for the first time the molecular targeting process of two 4-arylthiosemicarbazides with imidazole substitutions in the N1 position in *T. gondii*. This study was based on reports in which the inhibitory activity of thiosemicarbazide and thiosemicarbazones against TYR, which catalyzes two initial sequential oxidations of Tyr to L-DOPA, was shown [40,41,42,43]. A new class of TYR inhibitors was also considered by our group. To confirm the concept that 4-arylthiosemicarbazide may act as an inhibitor of the *T. gondii* enzymes involved in Tyr metabolism, the influences of the most potent anti-TYR compounds with *para*-nitro and *meta*-iodo substitutions, **2a** and **6a**, respectively [38,39], during *T. gondii* growth into human foreskin fibroblast cells were examined compared to the reference (TYR is an inhibitor of kojic acid) [16]. These compounds were selected from 58 new 4-arylthiosemicarbazide derivatives that were tested for their activity and selectivity against *T. gondii*. Our hypothesis was also indirectly confirmed by Marino and Boothroyd, who observed a decrease in the number of parasitophorous vacuoles per cell and a decrease in the average number of doubled tachyzoites post-infection in medium without Tyr, which is related to the fact that *Toxoplasma* tachyzoites require exogenous Tyr for growth [12]. Amino acid auxotrophy in *T. gondii* tachyzoites or bradyzoites has been a long recurring research subject [10,12], and like many unicellular pathogens, *T. gondii* requires exogenous Phe and Tyr [9]. The enzymes are crucial in the conversion of Phe to Tyr and then to L-DOPA and are both aromatic amino acid hydroxylases (AAHs), which are present in the tachyzoite or bradyzoite. Interestingly, higher levels of AAHs are expressed during bradyzoite differentiation, the stage with low metabolic activity [12,14]. Additionally, it was described that AAH2 plays a moderate role, while AAH1 shows a much stronger role in the formation of oocysts during infection in cats. Moreover, AAH1 plays a role in parasite survival inside the cat intestinal epithelium at early stages of merogony and schizogony [13]. Thus, AAHs are important at all stages of the life cycle, and *Toxoplasma* are dependent on the addition of exogenous Tyr for efficient growth [12].

Since our previous experiments with TYR and 4-aryltiosemicarbazides [16] did not provide direct evidence that the tested compounds inhibit the activity of the aromatic amino acid hydroxylases, in this work, both enzymatically active proteins were obtained in the *E. coli* expression system and used for further investigation. AAH1 is detectable in all stages, whereas AAH2 expression is induced during bradyzoite differentiation, so in our study, we converted tachyzoites of the ME49 strain to bradyzoites to obtain both AAHs. We proved that compounds **2a** and **6b** with thiosemicarbazide scaffolds demonstrated inhibitory effects on both aromatic amino acid hydroxylase activities. Detailed kinetic studies showed the competitive character of inhibition on AAHs, and a lower inhibitory constant was observed for both derivatives against AAH1. Additionally, AAH inhibitors were explored in a drug affinity responsive target stability assay, which depends on the concept that ligand-bound proteins demonstrate modified susceptibility to enzymatic degradation relative to unbound proteins [44]. In kinetic studies, it was observed that inhibitory constants for **2a** and **6b** were lower in relation to AAH1; therefore, only this hydroxylase was used for this assay. Both studied compounds significantly inhibited proteolysis of AAH1 by protease, and a dose-dependent enhancement was observed in the level of AAH1. These results clearly show that both of our derivatives bind to the toxoplasmic aromatic amino acid hydroxylase.

The very important role of AAH1 was also indicated by Zhang et al., who reported that AAH1 was expressed and located on the surface of *T. gondii* tachyzoites and could be used as a vaccine candidate antigen to mediate cell-mediated and humoral immunity. BALB/c mice were immunized with AAH1, and significant increases in IgG, IgG2a, IgG1, IFN-γ, IL-4, and IL17 production were observed. Additionally, AAH1-vaccinated animals apparently had a prolonged survival time after infection with the RH strain of *T. gondii* compared to that of nonvaccinated animals [17]. Moreover, other *T. gondii* proteins, e.g., dense granules (GRA), during invasion can activate immune responses through the activation of NF-κB [37,45,46]. In our study, we observed that soluble lysate of tachyzoites or bradyzoites significantly activated the NF-κB pathway in human monocytes, and as a result, INF-γ was produced. Furthermore, similar activation was observed after stimulation with both catalytic domains of AAHs.

The NF-κB family consists of proteins that associate with each other to form homo or heterodimeric complexes. NF-κB molecules are retained in the cytoplasm by interaction with kappa-B (IκB) inhibitors. In response to stimuli, such as LPS or other *T. gondii* proteins, IκB is phosphorylated by IκB kinase (IKK) and degraded by the ubiquitin–proteasome. Degradation of IκB released NF-κB and enabled its translocation into the nucleus to activate multiple inflammatory-associated genes [36,37]. Shawish et al. showed that thiosemicarbazone derivatives of 2,3-dihydroxybenzaldehyde, which are formed from thiosemicarbazides, have anti-inflammatory effects in vitro and in vivo [36]. Therefore, Nazim et al. synthesized the 1-indanone thiosemicarbazone by molecular hybridization of 1-indanone and thiosemicarbazide and examined its anti-inflammatory activity. The results showed that these derivatives exhibited remarkable anti-inflammatory activity [34]. The described molecular mechanism of thiosemicarbazone derivatives on the NF-κB pathway is related to the inhibition of IκB degradation, which in turn blocks the nuclear translocation of molecules and downregulates the expression of NF-κB target genes. A molecular docking study showed a potential interaction between compounds and the active site of IKK, which could explain the mechanism of inhibition of NF-κB activation [33,34].

In our study, we used THP1-Blue™ reporter cells derived from the human THP-1 monocyte cell line by stable integration of an NF-κB-inducible secreted embryonic alkaline phosphatase (SEAP) reporter construct. As a result, activating NF-κB pathways releases SEAP into the culture supernatant, and the level can be measured spectrophotometrically using SEAP detection reagent. We observed that both derivatives significantly block NF-κB pathways in human monocytes stimulated with toxoplasmic soluble antigens and AAHs or LPS, which is correlated with a significant decrease in IFN-γ production in treated monocytes. Thus, there is a high probability that our compounds with thiosemicarbazide scaffolds, beyond AHH inhibition, were also capable of inhibiting the active site of IKK, similarly to derivatives with thiosemicarbazone scaffolds. Moreover, the anti-inflammatory activity of the tested 4-arylthiosemicarbazide derivatives could be strategic in limiting the development of the acute phase of toxoplasmosis, but also could reduce the level of IFN-γ production, which prolongs the host’s liveliness by reducing the cytokine storm [47,48,49].

## 4. Materials and Methods

### 4.1. Chemistry and Compound Preparation

All 4-arylthiosemicarbazide derivatives were synthesized by our research group previously. The methods of synthesis and physicochemical characteristics details of Compounds **2a** and **6a** were presented in our previously published papers [16,32,38,39].

Compounds **2a** and **6a** were dissolved in dimethyl sulfoxide (DMSO, Sigma–Aldrich, St. Louis, MO, USA) to 1 M. The final concentration of DMSO in the compound dilutions in all tests was not higher than 0.1%. All compounds were freshly prepared before the experiment and sterile filtered.

### 4.2. Cell Culture

Hs27 cells (human fibroblast, ATCC^®^ CRL-1634™, Manassas, VA, USA) were cultured in DMEM (Dulbecco’s Modified Eagle’s Medium, ATCC^®^ 30-2002™) supplemented with 10% fetal bovine serum (FBS, ATCC^®^ 30-2020™), 100 IU/mL penicillin and 100 μg/mL streptomycin (ATCC^®^ 30-2300™). Cells were trypsinized (Trypsin-EDTA Solution, ATCC^®^ 30-2101™) twice a week, seeded at a density of 1 × 10^6^ per T25 cell culture flask (Corning^®^, Sigma–Aldrich) and incubated at 37 °C and 5% CO_2_ to achieve a confluent monolayer. THP1-Blue™ NF-κB reporter cells (monocytes, Invivogen, Toulouse, France) were cultured in RPMI 1640 (22400089, Gibco™, Thermo Scientific, Waltham, MA, USA), 10% heat-inactivated fetal bovine serum, 100 μg/mL Normocin™ (Invivogen, Toulouse, France), 10 μg/mL blasticidin (Invivogen, Toulouse, France), 100 IU/mL penicillin and 100 μg/mL streptomycin. Cells were incubated at 37 °C and 10% CO_2_ and passaged every 5 days to maintain a density of at least 2 × 10^6^ cells/mL.

### 4.3. Parasite Culture

The ME49 strain of *Toxoplasma gondii* (ATCC^®^ 50611™, second haplogroup) was maintained as tachyzoites in parasite culture medium, which contains DMEM with 3% HIFBS (heat-inactivated FBS; 1 h at 56 °C). Infected cells were incubated at 37 °C and 5% CO_2_. Bradyzoites were induced in vitro using the high (pH 8.2) pH shock method [50]. Bradyzoite differentiation medium contained high glucose DMEM powder (D5648, Sigma–Aldrich), 50 mM HEPES (H3375, Sigma–Aldrich), and 1% FBS. Media were adjusted to pH 8.2 with freshly made 1 M NaOH (S8045, Sigma–Aldrich) and sterilized by filtration. The medium was replaced every 1–2 days with fresh differentiation medium, which reduces the likelihood of bradyzoites reverting to tachyzoites. After 7 days, differentiated bradyzoites were confirmed with immunofluorescence assays using rabbit MAG1 antibodies as bradyzoite-specific markers and rabbit SAG1 antibodies as tachyzoite-specific markers (see Section 4.4 and Section 4.5). Six parasitized monolayers were harvested after 7 days by scraping, and bradyzoites were purified by passage through a 27 gauge needle, washed in phosphate-buffered saline (PBS) and filtered through a 0.45 μm filter. Parasites were enumerated, and pellets were resuspended in phenozol for RNA extraction using a Total RNA Mini and Clean-Up RNA Concentrator (A&A Biotechnology, Gdańsk, Poland).

### 4.4. Development of Anti-SAG1 and Anti-MAG1 Rabbit Polyclonal Antibodies

Immunization of laboratory New Zealand rabbits with recombinant SAG1 and MAG1 toxoplasmal proteins was performed to prepare antigen-specific sera. The laboratory animals were raised under standard conventional conditions that were approved by the Polish Ministry of Science and Higher Education animal facility of the Institute Microbiology, Biotechnology and Immunology, Faculty of Biology and Environmental Protection, University of Lodz. The experimental procedures were approved and conducted according to the guidelines of the appropriate Polish Local Ethics Commission for Experiments on Animals No. 9 in Lodz (Agreement 41/ŁB343/2006).

Recombinant surface protein SAG1 and tissue cysts/bradyzoite marker protein MAG1 were expressed and purified as we previously described [51,52]. Briefly, *T. gondii* total DNA and standard molecular biology procedures were employed for the gene cloning strategies [53]. The 882 bp fragment of SAG1 (S76248), encoding 294 amino acids (aa 18-311), and 1272 bp fragment of MAG1 (U09029) encoding 424 amino acids (aa 33-452) were amplified by PCR and finally cloned into the pHis expression vector using EcoRI and Hind III or BamHI and HindIII restriction enzymes, respectively. The resultant vectors were introduced into Escherichia coli BL21 (DE3) cells for further protein expression. The recombinant SAG1 and MAG1 proteins contained a 6-His tag within an extra 30 aa and 28 aa, respectively, and were successfully purified by affinity Ni^2+^-chromatography (HisPur™ Ni-NTA Resin, Thermo Scientific, Waltham, MA, USA).

New Zealand rabbits were immunized subcutaneously with three doses of recombinant SAG1 or MAG1 (dose I-200 µg; doses II and III-150 µg in 0.4 mL of PBS) emulsified with an equal volume of incomplete Freund’s adjuvant (F5506, Sigma). The vaccinations were performed in 3-week intervals, and 7 days after the last booster, blood samples were collected to estimate the immunization-promoted generation of serum anti-SAG1 and anti-MAG1 IgG antibodies. The levels of protein-specific immunoglobulins were detected with an indirect ELISA using rabbit sera (twofold dilutions ranging from 1/200 to 1/102.400), horseradish peroxidase (HRP)-labeled goat IgG anti-rabbit IgG immunoglobulins (Jackson ImmunoResearch, Hongkong, China) (dilution 1/2000), and ABTS (2,2’-azino-bis(3-ethylbenzothiazoline-6-sulfonic acid) (Sigma) as the primary antibodies, secondary antibodies, and chromogen, respectively. Absorbance values were measured at λ = 405 nm using a Multiscan EX ELISA reader (Thermo Scientific, Waltham, MA, USA). The optimal dilution of secondary antibodies was determined in the preliminary titration assays. After establishing the immunization efficacy, the laboratory rabbits were euthanized, and blood was collected for serum preparation.

### 4.5. Immunofluorescence Staining of Tachyzoites/Bradyzoites

After 7 days, differentiated bradyzoites were confirmed with immunofluorescence assays using rabbit MAG1 antibodies as bradyzoite-specific markers and rabbit SAG1 antibodies as tachyzoite-specific markers. Both cell monolayers of Hs27 with tachyzoites or bradyzoites were washed with PBS and fixed with a formaldehyde solution (252549, Sigma–Aldrich) of 3.7% in PBS for 20 min. Next, the cells were blocked for 30 min with 10% FBS in PBS and permeabilized with 0.1% Triton-X100 for 15 min. The following monolayers were washed with PBS, and serum with anti-SAG1 or anti-MAG1 antibodies (1:500) was added for 30 min at 37 °C. Then, the cells were washed with PBS, and FITC-labeled goat anti-rabbit IgG (Bio–Rad Antibodies, Hercules, CA, USA) 1:100 was added for 30 min at 37 °C. Finally, the cells were washed with PBS, and tachyzoites or bradyzoites were visualized with a fluorescence inverted microscope at a magnification ×200 (IX50 microscope with UC90 camera, Olympus, Tokyo, Japan) using CellSens software (CellSens Standard 2.3, Olympus).

### 4.6. Expression and Purification of Recombinant AAHs

In this study, the catalytic domains of AAH1 (TGME49_287510) and AAH2 (TGME49_212740) were obtained. Two sets of primers were used to amplify the *aah1 and aah2* genes (AAH1—forward primer, 5’ CCTGCAGCGAAGGCTGTCGATCAATAACGTT; AAH1—reverse primer, 5’ CAAGCTTCTAGAACCTGAGGGAAACGGGC; AAH2—forward primer, 5’ CCTGCAGCGAAGGCTGTCGATCAATAACGTT; AAH2—reverse primer, 5’ CAAGCTTCTAGATCTTGAGGGAGACAGGAGGC). The amplified products were cloned into the pJET 1.2 vector (Thermo Scientific), digested with Pst I and Hind III restriction enzymes and inserted into the pET 45b expression vector (Novagen, Madison, WI, USA). The obtained AAH1-pET45b and AAH2-pET45b constructs were electroporated into the BL21 GOLD (DE3) pLysS *E. coli* strain (Novagen). For the expression of recombinant AAH1 and AAH2 proteins, the cultures were grown in Terrific Broth medium (yeast extract 24 g/L, tryptone 20 g/L, glycerol 4 mL/L, and 100 mL 1 M phosphate buffer—all reagents from Sigma–Aldrich) supplemented with 10 mM ferrous ammonium sulfate (F1543, Sigma–Aldrich) at 37 °C to an OD_600_ of 0.8 and induced with 1 mM IPTG (I6758, Sigma–Aldrich) for 20 h at 18 °C. Cultures were harvested by centrifugation at 10,000× *g* for 30 min at 4 °C, and the pellets were resuspended in cold lysis buffer as follows: 50 mM Tris-HCl pH 7.5, 150 mM NaCl (S9888, Sigma–Aldrich), 100 mM imidazole (56750, Sigma–Aldrich), 10% glycerol (G5516, Sigma–Aldrich), 0.2% Triton-X 100 (93443, Sigma–Aldrich), 0.1 M PMSF (Roche, Nutley, NJ, USA), 10 mg/mL lysozyme (Sigma–Aldrich), 10 kU/mL DENARASE^®^ (c-LEcta, Leipzig, Germany), 0.14 μL/mL 2-mercaptoethanol (M3148, Sigma–Aldrich), and EDTA-free protease inhibitors (Roche). Then, the samples were sonicated for 30 s with 30 s rest eight times on ice and centrifuged at 21,000× *g* for 60 min at 4 °C. The resulting soluble fraction was filtered through a 0.45 μm filter, purified by affinity Co^2+^-chromatography (HisPur™ Cobalt Resin, Thermo Scientific) and eluted with the following cold elution buffer: 50 mM Tris-HCl pH 7.5, 150 mM NaCl, 500 mM imidazole, and 10% glycerol. The imidazole was removed by a purification system with a 10 kDa Amicon^®^ Ultra0.5 device (Millipore) for 2 h rs at 4 °C and replaced with 50 mM Tris-HCl pH 7.5 plus 0.2% Triton-X100. SDS–PAGE electrophoresis and Western blotting with anti-His-tag (SAB1306082, Sigma–Aldrich) and polyclonal anti-tyrosine hydroxylase (anti-TH) antibodies (AB152, Sigma–Aldrich) were used to assess the purity of each protein.

### 4.7. Biochemical Assays

Kinetic analyses of AAH1 and AAH2 were performed by measuring the oxidation of tetrahydrobiopterin ((6R)-BH_4_, T4425, Sigma–Aldrich) using a coupled reaction measuring the decrease in absorbance at 340 nm due to NADH oxidation, as previously described [14]. The protein concentration was determined using the Bradford reagent (B6916, Sigma–Aldrich). All assays were initiated by the addition of 10 μg enzyme. Standard conditions were pH 7.5, 50 mM HEPES, 100 μg/mL catalase (C100, Sigma–Aldrich), 500 μM NADH (N8129, Sigma–Aldrich), 10 μM ferrous ammonium sulfate, 0.25 I.U./mL sheep liver dihydropteridine reductase (D6888, Sigma–Aldrich), and 500 μM (6R)-BH_4_. Varied concentrations of 0–0.2 mM L-tyrosine (93829, Sigma–Aldrich) were used. Assays were performed at 32 °C. The decrease in absorbance was determined spectrophotometrically using the multimode microplate reader SpectraMaxVR i3 (Syngen).

The inhibition kinetic type was evaluated by determining the intersection point in the Lineweaver–Burk plots. The results from three experiments are shown. All the data were fit into the following competitive inhibition equation to obtain the *K_I_* value:$$v=\frac{V_{max}}{1+\frac{K_{M}}{\left[S\right]}\left(1+\frac{\left[I\right]}{K_{I}}\right)}$$
where *v* is the velocity of reaction, *V_max_* is the maximal velocity of reaction, *K_M_* is the Michaelis constant, *K_I_* is the inhibitory constant, [*S*] is the enzyme substrate concentration, and [*I*] is the inhibitor (compound) concentration.

### 4.8. Drug Affinity Responsive Target Stability Assay

Drug affinity responsive target stability (DARTS) is a robust method for the detection of novel small molecule protein targets. DARTS is based on the stabilization of the target protein that occurs upon small molecule binding by detecting the binding-induced increase in resistance to proteolysis [44,54]. The AAH1 concentration was determined using the Bradford reagent and diluted in TNC buffer to 2.5 μg/μL. TNC buffer (1 mL, 10×) 500 μL 1 M Tris-HCl pH 8.0, 100 μL 5 M NaCl, 100 μL 1 M CaCl_2_ (C1016, Sigma–Aldrich), and 300 μL ultrapure water.

#### 4.8.1. Protection of AAH1 from Proteolysis by 4-Arylthiosemicarbazide Derivatives in the Presence of 10 µM Compound

To 99 μL of AAH1, 1 μL of 1 mM **2a** or **6a** was added. As a control, 1 μL of DMSO (vehicle control) was used. Samples containing the protein and compound were incubated for 60 min at room temperature with shaking. Subsequently, Pronase (10 mg/mL, Roche) dilutions were prepared (1:100, 1:200, 1:400, and 1:800) in TNC buffer. After incubation with 4-arylthiosemicarbazide derivative or DMSO, each sample was split into 20 μL, and 2 μL of the range of Pronase solutions in each sample was added. For the nondigested sample, 2 μL of 1× TNC buffer instead of protease was used. Samples were incubated at room temperature with protease for 15 min. The digestion reaction was stopped by adding 2 μL of 20× EDTA-free protease inhibitors and incubating on ice for 10 min. Then, SDS–PAGE electrophoresis and Western blotting with polyclonal anti-TH antibody were performed.

#### 4.8.2. Protection of AAH1 from Proteolysis by 4-Arylthiosemicarbazide Derivatives in the Presence of Increasing Concentrations of Compound

To 20 μL of AAH1, 0.2 μL of various concentrations of 0–10 mM **2a** or **6a** was added. Samples containing the protein with compound were incubated for 60 min at room temperature with shaking. After incubation, 2 μL of the 1:400 Pronase solution was added to each sample. For the nondigested sample, 2 μL of 1× TNC buffer instead of protease was used. Samples were incubated at room temperature with protease for 15 min. The digestion reaction was stopped by adding 2 μL of 20× EDTA-free protease inhibitors and incubating on ice for 10 min. Then, SDS–PAGE electrophoresis and Western blotting with polyclonal anti-TH antibody were performed. We also performed this assay with bovine serum albumin (BSA, A2153) as a control to exclude the inhibition of Pronase by our compounds.

### 4.9. Preparation of Soluble Tachyzoite and Bradyzoite Antigens

Soluble tachyzoite (TLA) and bradyzoite (BLA) antigens were prepared from tachyzoites and bradyzoites of the *T. gondii* ME49 strain, as previously described with modification [44]. Briefly, tachyzoites and bradyzoites were washed twice with cold PBS. Afterward, 1 × 10^9^ cells were lysed for 10 min on ice in 500 μL of cold lysis buffer containing 345 μL of M-PER cell lysis reagent (78501, Thermo Fisher), 25 μL of 20× protease inhibitor cocktail (Roche), 25 μL of 1 M sodium fluoride (S7902, Sigma–Aldrich), 50 μL of 100 mM β-glycerophosphate (G9422, Sigma–Aldrich), 50 μL of 50 mM sodium pyrophosphate (221368, Sigma–Aldrich), and 5 μL of 200 mM sodium orthovanadate (450243, Sigma–Aldrich). Then, the samples were centrifuged at 18,000× *g* for 15 min at 4 °C. The supernatants were collected and filtered for sterilization, and the final TLA and BLA were stored at −70 °C. The concentration of soluble antigens was determined by the Bradford method with the BSA standard.

### 4.10. Quantification of NF-κB Induction

To address the whether the 4-arylthiosemicarbazide derivatives have anti-inflammatory activity, we performed a screening assay on THP1-Blue™ NF-κB reporter cells, and this assay enables the induction of the NF-κB transcription factor to be quantified. Monocytes were stimulated with *T. gondii* antigens, TLA, BLA, AAH1, or AAH2 at a concentration of 10 μg. Additionally, for these assays, we removed endotoxin from the recombinant proteins AAH1 and AAH2 with Triton™ X-114 (X-114, Sigma–Aldrich) phase separation, as previously described [55]. THP1-Blue™ NF-κB cells were seeded in 96-well plates at a density of 1 × 10^5^ cells/well. Afterward, two approaches were used. In the first approach, monocytes were treated with compounds **2a** or **6a** at a concentration of 10 μM for 2 h, and then stimulants were added for another 22 h; in the second approach, monocytes were first stimulated for 2 h and then treated for another 22 h with compound **2a** or **6a** at a concentration of 10 μM. In total, the cells were incubated for 24 h. Next, 20 μL of supernatant from the monocyte cultures was added to 200 μL of QUANTI-Blue™ Solution (Invivogen) reagent, and the quantification of alkaline phosphatase, as a marker of cell activation (inflammatory response), was measured at 650 nm. The following internal controls were included in the test: untreated monocyte cultures served as negative controls, and monocytes treated only with **2a** or **6a** compounds (data not shown) and cultures treated with *E. coli* lipopolysaccharide (LPS, L3024, Sigma–Aldrich) at a concentration equal to 0.25 I.U./mL were positive controls. Experiments were independently repeated three times.

### 4.11. Cytokine Determination

The concentrations of interferon gamma (IFN-γ) in the supernatants of stimulated THP1-Blue™ NF-κB cells were determined with commercially available OptEIA™ ELISA sets (BD Biosciences, San Jose, CA, USA) according to the manufacturer’s instructions. The analyses were conducted based on data obtained from 3 individual experiments.

### 4.12. Data Analysis

Statistical analyses and graph making were performed using GraphPad Prism version 9.0.0 on macOS (GraphPad Software, San Diego, CA, USA). Kinetic data were first fitted to the Michaelis–Menten equation and then linearized to Lineweaver–Burk plots using GraphPad Prism. The intensity of the Western blot bands was quantified using ImageJ software (National Institutes of Health, Bethesda, MD, USA). To assess the significance of differences, one-way ANOVA was conducted, and for significant comparisons, further analysis was performed using Tukey’s multiple comparisons. Additionally, two-way ANOVA was conducted, and for significant comparisons, further analysis was performed using Dunnett’s multiple comparisons test. The differences were considered significant with *p* values < 0.05.

## 5. Conclusions

Collectively, the obtained results support the conclusion that the examined compounds with thiosemicarbazide scaffolds are adapted to disrupt Tyr, and probably Phe, metabolism in *Toxoplasma* tachyzoites or bradyzoites by deregulating aromatic amino acid hydroxylases. Intriguingly, the wide biological activity of thiosemicarbazide derivatives was confirmed again because our compounds also possess anti-inflammatory activity by blocking NF-κB pathways in human monocytes.

## Figures and Tables

**Figure 1 ijms-23-03213-f001:**
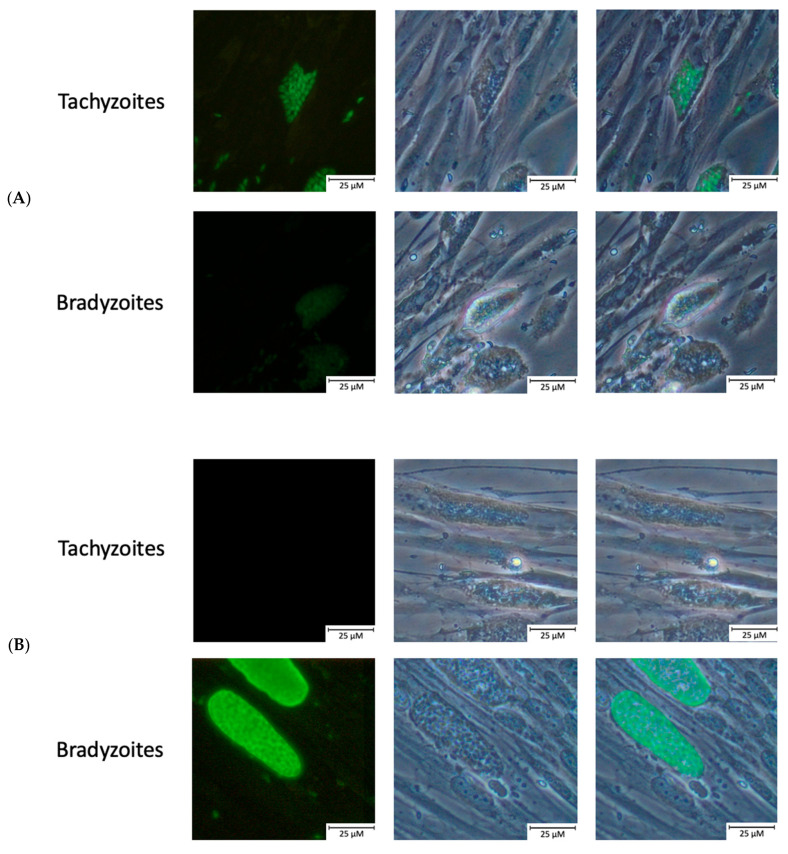
Confirmation of the differentiation of tachyzoites to bradyzoites by an immunofluorescent assay. (**A**) SAG1 tachyzoite-specific marker expression in *Toxoplasma gondii* ME49 strain. (**B**) MAG1 bradyzoite-specific marker expression in *Toxoplasma gondii* ME49 strain.

**Figure 2 ijms-23-03213-f002:**
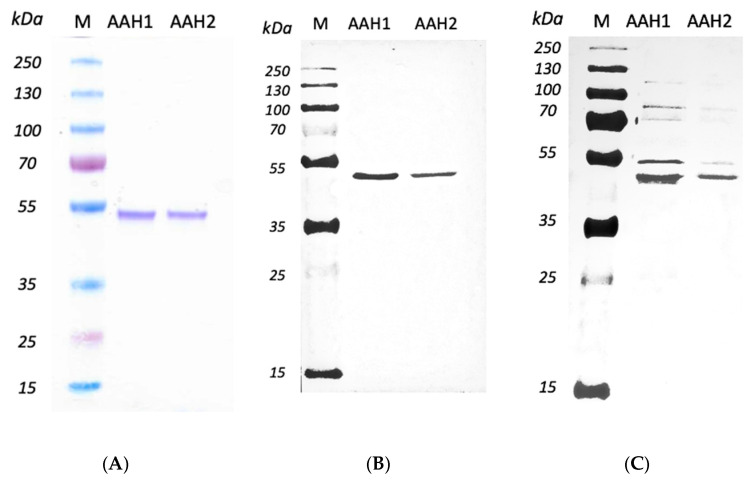
*T. gondii* aromatic amino acid hydroxylases expressed in this study. SDS–PAGE and Western blots of proteins expressed from AAH1 and AAH2. (**A**) SDS–PAGE of AAH1 and AAH2. (**B**) Monoclonal antibody anti-6xHis Tag. (**C**) Polyclonal antibody anti-TH. Due to the high degree of similarity with TH, recombinant toxoplasmic proteins are recognized by polyclonal antibodies against TH.

**Figure 3 ijms-23-03213-f003:**
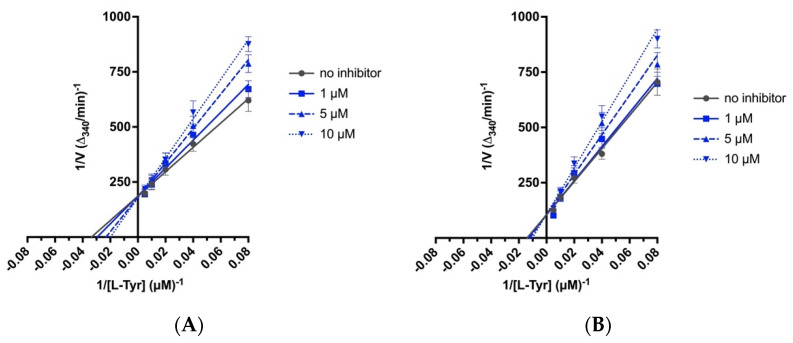
Lineweaver–Burk linearized plots with varying concentrations of compounds **2a** and **6a**. The concentration of inhibitor varied from 0 to 10 μM. As a substrate, L-tyrosine (L-Tyr) was used with 0.125, 0.25, 0.5, 1, and 2 mM concentrations. V—reaction rate. (**A**) AAH1 and **2a** as inhibitors. (**B**) AAH2 and **2a** as inhibitors. (**C**) AAH1 and **6a** as inhibitors. (**D**) AAH2 and **6a** as inhibitors. Each point represents the combined averages of three independent experiments in triplicate under the same conditions.

**Figure 4 ijms-23-03213-f004:**
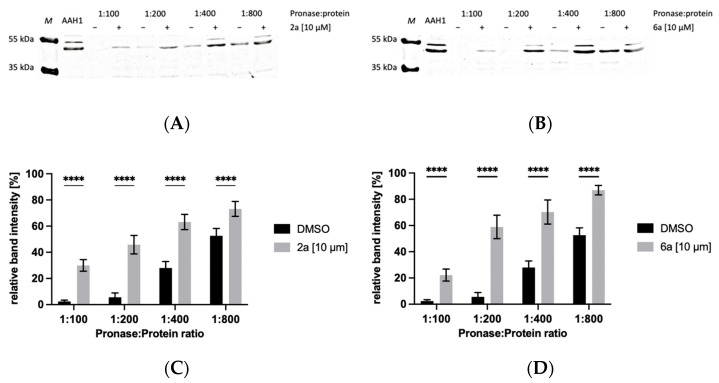
Protection of AAH1 from proteolysis by 4-arylthiosemicarbazide derivatives in the presence of 10 µM compound and with the Pronase:protein ratio. Western blots of AAH1 from the drug affinity responsive target stability assay and relative band intensity analysis. (**A**) Protection of AAH1 from proteolysis by **2a** in the presence of 10 µM compound. (**B**) Protection of AAH1 from proteolysis by **6a** in the presence of 10 µM compound. (**C**) Proteolytic activity inhibition by **2a** protecting AAH1. (**D**) Proteolytic activity inhibition by **6a** protecting AAH1. The intensities of the AAH1 bands were quantified using ImageJ software. Values with statistically significant differences are labeled by asterisks; **** *p* < 0.0001. Data were compared using one-way ANOVA with Tukey’s multiple comparisons test. Data were obtained from three independent experiments and are expressed as means ± standard deviations.

**Figure 5 ijms-23-03213-f005:**
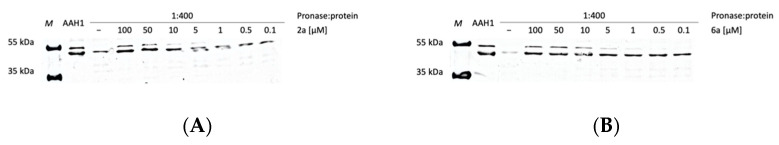
Illustration of the amount of stabilized AAH1 accessible for detection in the presence of increasing concentrations of 4-arylthiosemicarbazide derivatives. (**A**) The stabilization effect of **2a** on AAH1 was evaluated by Western blot. (**B**) The stabilization effect of **6a** on AAH1 was evaluated by Western blot. (**C**) Dose-dependence curve of **2a** on AAH1. (**D**) Dose-dependence curve of **6a** on AAH1. The intensities of the AAH1 bands were quantified using ImageJ software. The line was fitted with a three-parameter logistic curve. Data were obtained from three independent experiments and are expressed as the means ± SD.

**Figure 6 ijms-23-03213-f006:**
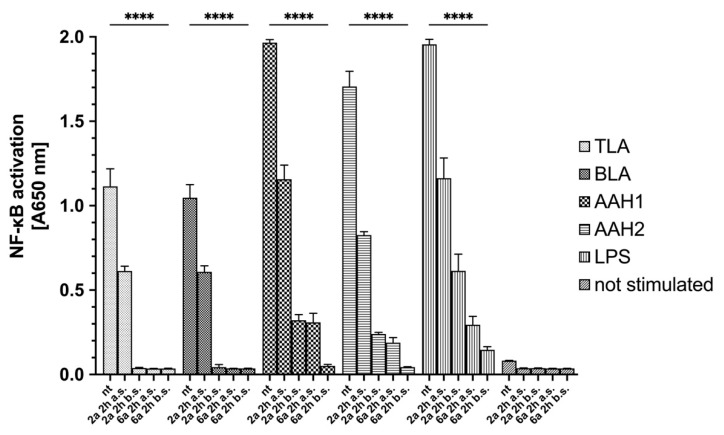
NF-κB induction in THP1-Blue™ human monocytes exposed to TLA, BLA, AAH1, or AAH2, or treated with LPS (positive control). Data are shown as the means from three repeats with error bars that indicate the standard deviations. Data were compared using a two-way ANOVA test, and for significant comparisons, further analysis was performed using Dunnett’s multiple comparisons test. Values with statistically significant differences are labeled by asterisks; **** means *p* < 0.0001. The differences were considered significant with a *p* value < 0.05 and are presented in Appendix A. Aberration: nt—not treated; a.s. after stimulation; b.s. before stimulation; TLA—soluble tachyzoite antigens; BLA—soluble bradyzoite antigens; AAH—aromatic amino acid hydroxylases.

**Figure 7 ijms-23-03213-f007:**
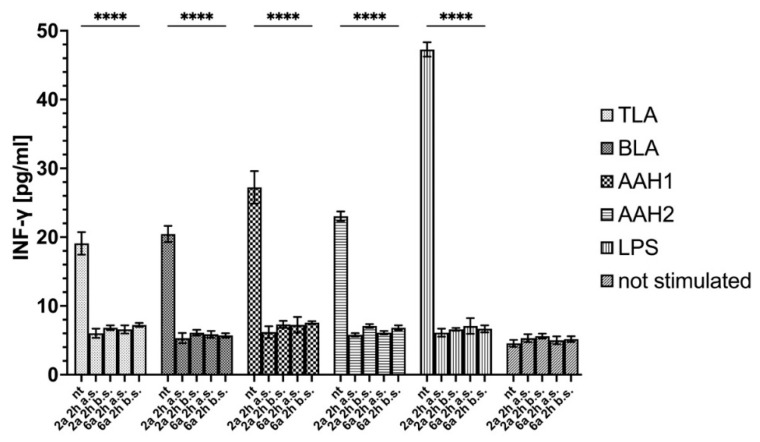
INF-γ production from supernatants after human monocytes were exposed to TLA, BLA, AAH1, or AAH2, or treated with LPS (positive control). Data are shown as the means from three repeats with error bars that indicate the standard deviations. Data were compared using a two-way ANOVA test, and for significant comparisons, further analysis was performed using Dunnett’s multiple comparisons test. Values with statistically significant differences are labeled by asterisks; **** *p* < 0.0001. The differences were considered significant when *p* values were less than 0.05, and such differences are presented in Appendix A. Aberration: nt—not treated; a.s. after stimulation; b.s. before stimulation; TLA–soluble tachyzoite antigens; BLA—soluble bradyzoite antigens; AAH—aromatic amino acid hydroxylases.

**Table 1 ijms-23-03213-t001:** The inhibitory concentrations (IC_50_) against *Toxoplasma gondii* and cytotoxic concentrations (CC_30_) against L929 and Hs27 cell lines of **2a** and **6a**.

Compound	IC_50_ ^a^	CC_30_ ^b,c^	CC_30_ ^b,d^	SR ^e^	K_I*TYR*_ ^f^
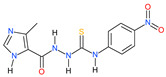 **2a**	67.49	682.98	697.67	10.34	9.35
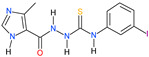 **6a**	25.67	213.27	209.45	8.16	12.4

^a^ IC_50_ (µM) represents the concentration of the compound necessary for 50% inhibition of *T. gondii* proliferation in vitro. IC_50_ values were determined based on the plotted curves. ^b^ CC_30_ (µM) represents the noncytotoxic concentration of the compound necessary for 30% cell proliferation inhibition in vitro. CC_30_ values were determined based on the plotted curves. ^c^ CC_30_ (µM) against the L929 cell line. ^d^ CC_30_ (µM) against the Hs27 cell line. ^e^ SR represents selectivity ratio values—each calculated as the ratio of 30% cytotoxic concentration (CC_30_) against Hs27 to 50% antiparasitic concentration (IC_50_). ^f^ K_I*TYR*_ (µM) represents the inhibitory constant, the concentration required to produce half maximum inhibition.

**Table 2 ijms-23-03213-t002:** Kinetic parameters of the AAH1 and AAH2 inhibition assays of compounds **2a** and **6a**.

Enzyme	Inhibitor	*V_max_*/*K_M_* ^a^	*K_M_* ^b^	*K_I_* ^c^
AAH1	none	0.1817	0.030	-
	**2a**	0.1272	0.042	1.487
	**6a**	0.1282	0.042	1.265
AAH2	none	0.1311	0.071	-
	**2a**	0.1094	0.085	3.691
	**6a**	0.1038	0.089	3.227

^a^*V_max_*/*K_M_* [min.^−1^] represents the catalytic efficiency as a ratio of the maximum rate of reaction (*V_max_*) to the Michaelis constant (*K_m_*). ^b^
*K_M_* [mM] represents the Michaelis–Menten constant, the substrate concentration at which the reaction rate was 50% of the maximum rate of reaction when all enzyme active sites were saturated with substrate (*V_max_*). ^c^
*K_I_* (µM) represents the inhibitory constant, the compound concentration required to produce half-maximum inhibition.

## Data Availability

Data are contained within the article or Appendix A.

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
