# Peer review of "4-Arylthiosemicarbazide Derivatives as Toxoplasmic Aromatic Amino Acid Hydroxylase Inhibitors and Anti-inflammatory Agents"

_ijms, 2022, doi:10.3390/ijms23063213_

Round 1

Author Response

Please find the respond in attachment.

Reviewer 2 Report

The manuscript of Bekier et al. describes the study of the potential mechanism of action of two 4-arylthiosemicarbazide derivatives, described in a previous paper. Numerous biological assays are described and commented to draw conclusions on different biological activities against Toxoplasma gondii.

The work is interesting enough, but it is the continuation and deepening of a previous paper.

Typographical errors:

line 112: not "4-arylothiosemicarbazide" but "arylthiosemicarbazide".

line 306: not “T. godnii "but" T. gondii ".

line 311: not "with Compounds" but "with compounds".

line 367: not “T. godnii "but" T. gondii".

line 389: not "of Compounds" but "of compounds".

line 510: not "4-arylothiosemicarbazide" but "arylthiosemicarbazide".

line 516: not "that Compounds" but "that compounds".

line 542: not “T. godnii "but" T. gondii " line 545: not "4-arylothiosemicarbazide" but "arylthiosemicarbazide".

line 31: what does "similarity of thiosemicarbazone scaffolds to thiosemicarbazones" mean ?

It is advisable to use the mL symbol instead of mL.

Check the legend of figure 3: (A, AAH1 and 2a; B, AAH2 and 6a; C, AAH1 and 2a; D, AAH2 and 6a).

References to figures 6 and 7 are missing from the text.

At several places, sentences illustrating properties of compounds are inserted, but the data is not shown. In such cases it is advisable not to include these sentences if they are not strictly necessary.

The final discussion highlights the capacity of compounds to disrupt Tyr metabolism in Toxoplasma, but also the anti-inflammatory activity mediated by blocking NF-kB pathway in human monocytes. It is not clearly discussed whether the presence of these two actions is positive and why.

Provided that these issues are settled the paper is worthy of pubblication.

Author Response

Please find the respond in attachment.
